# E4orf1 Prevents Progression of Fatty Liver Disease in Mice on High Fat Diet

**DOI:** 10.3390/ijms23169286

**Published:** 2022-08-18

**Authors:** Rownock Afruza, Nikhil V. Dhurandhar, Vijay Hegde

**Affiliations:** Obesity and Metabolic Health Laboratory, Department of Nutritional Sciences, Texas Tech University, Lubbock, TX 79409, USA

**Keywords:** obesity, NAFLD, NASH, E4orf1, diet reversal

## Abstract

Non-alcoholic fatty liver disease (NAFLD) covers a broad spectrum of liver diseases ranging from steatosis to cirrhosis. There are limited data on prevention of hepatic steatosis or its progression to liver disease. Here, we tested if either transgenic (Tg) doxycycline-induced expression in adipose tissue of E4orf1 (E4), an adenoviral protein, or dietary fat restriction attenuated hepatic steatosis or its progression in mice. Twelve to fourteen-week-old TgE4 mice (E4 group) and control mice were exposed to a 60% (Kcal) high fat diet (HFD) for 20 weeks, while another group of mice on HFD for 10 weeks were switched to a chow diet (chow group) for another 10 weeks. Glycemic control was determined at weeks 10 and 20. Tissues were collected for gene and protein analysis at sacrifice. Compared to control, diet reversal significantly reduced body weight in the chow group, whereas E4 expression attenuated weight gain, despite HFD. E4 mice evinced significantly improved glucose clearance, lower endogenous insulin secretion, reduced serum triglycerides, attenuated hepatic steatosis and inflammation. Interestingly, in spite of weight loss and lower liver fat, chow mice showed significant upregulation of hepatic genes involved in lipid metabolism. Despite HFD, E4 prevents hepatic lipid accumulation and progression of hepatic steatosis, while diet reversal maintains hepatic health, but is unable to improve molecular changes.

## 1. Introduction

Obesity and related co-morbidities, such as diabetes, non-alcoholic liver disease (NAFLD) and cardiovascular disease, are some of the most common health concerns [1]. Excessive calorie intake is often associated with hyperglycemia, hyperinsulinemia, oversized fat pad and liver damage [2,3]. Fat storage-related liver diseases can range from steatosis (accumulation of excessive fat in hepatocytes), nonalcoholic steatohepatitis (NASH; steatosis along with hepatocellular ballooning, lobular inflammation, cell death and pericellular fibrosis), cirrhosis (development of regenerative nodules surrounded by fibrous bands and portal hypertension) to hepatocellular carcinoma (HCC) [4]. NASH is a significant risk factor for the development of cirrhosis and HCC [5,6,7]. The disease’s overwhelming of the ability of the liver to handle carbohydrates and fatty acids (FAs) is the signal sign of developing NAFLD and NASH [8,9,10,11,12]. Overload of metabolites from carbohydrates and the FAs pathway induces hepatocellular stress, injury and death, leading to fibrogenesis, cirrhosis and hepatocellular carcinoma in the presence of genomic instability. If there is an excess supply or impaired disposal of FA, lipotoxic metabolites provoke endoplasmic reticular stress and cell injury [13].

E4orf1 (E4) is a protein from the human adenovirus Ad36, which is responsible for the anti-hyperglycemic effect of the virus [14]. E4 works through the Ras-mediated phosphoinositol 3-kinase (PI3K) to increase cellular translocation of glucose transporter (GLUT) 4 and glucose uptake while bypassing the proximal insulin signaling and selectively enhancing the distal insulin signaling pathway. In vitro and in vivo studies collectively show the anti-hyperglycemic effect of E4 [15,16,17,18,19,20], even under conditions of impaired proximal insulin signaling. This insulin-independent glucose disposal by E4 reduces the requirement of endogenous insulin for glucose disposal, which in turn reduces hyperinsulinemia [14,16]. In hepatocytes, E4 reduces glucose output under basal and gluconeogenic conditions, reduces de-novo lipogenesis, increases complete FA oxidation, and promotes lipid export [21], which collectively suggest reduction in hepatic lipid storage. Further, in mice, reducing endogenous insulin by E4 protects against hepatic lipid accumulation [16,18,19,22]. Moreover, E4 prevents excessive renal lipid accumulation, lipotoxicity, inflammation and injury in mice [23,24].

Under HFD conditions, hyperinsulinemia and insulin resistance are two connective threads for NAFLD. The insulin sparing action of E4 can reduce high levels of circulating endogenous insulin and improve the glycemic control by activating the distal insulin signaling pathway independent of obesity, thereby improving NAFLD. But the ability of E4 to modulate existing hepatic steatosis, and whether its effects are comparable to those of dietary fat restriction, has not been investigated. Lifestyle modification and a change to a healthy diet are important steps in managing hepatic steatosis [25]. However, maintaining long term diet and lifestyle modification is challenging [26]. Therefore, in the current study, we investigated whether E4 can prevent progression of HFD-induced lipid accumulation in the liver, despite continued HF feeding and if so, whether this result is comparable with similar results obtained through high fat diet reversal.

## 2. Results

### 2.1. Switching to Chow Diet Reduces Body Weight, but Inducing E4 Expression Attenuates Weight Gain despite the HFD

To determine the effect of *E4orf1* gene expression or dietary fat restriction on existing hepatic steatosis, mice were placed on 60% HFD for 10 weeks before exposing them to either doxycycline-supplemented 60% HFD for an additional 10 weeks or doxycycline-free regular chow diet for another 10 weeks. Body weights of the two groups were compared with those of the control group, which also continued on HFD with doxycycline for an additional 10 weeks. As expected, protein expression analysis showed doxycycline induced *E4orf1* transgene expression in the E4 mice group, but not in the chow or the control groups (data not shown). There were no significant changes in weekly average body weights between control mice and E4 mice after 20 weeks of HFD, while chow mice showed significant reductions in weekly average body weight compared to control, once they were switched to the regular chow diet (Figure 1). Average baseline body weights were 27.1 g, 24.2 g and 28.3 g in the control, E4 and chow groups, respectively. After 10 weeks of high fat feeding average body weights were 40.2 g, 36.8 g and 42.6 g in the control, E4 and chow groups, respectively. The average weight gains were 13.0 ± 0.5 g, 12.7 ± 1.1 g and 14.4 ± 1.1 g after 10 weeks of high fat feeding in the control, E4 and chow groups, respectively. At the end of the study, the average weight gain (15.5 g) was very similar to the measurement post 10 weeks of HFD in control mice, while it was only 3.6 g in E4 mice. After a change of diet from HF to chow, mice, on average, were 7.7 g lighter, compared to measurements post 10 weeks of HFD (Figure 1).

### 2.2. E4 Expression Reduced Endogenous Insulin Response to Glucose Load, and Improved Glycemic Control

Glucose tolerance tests (GTT) were done by intraperitoneally injecting mice with 2.0 g/kg glucose at weeks 10 and 20. In terms of glucose clearance, there were no significant differences between glucose and insulin in the groups at week 10 (Appendix A) and at the end of the study week-20 (Figure 2a–d); however, E4 mice, but not chow mice, required significantly lower amounts of endogenous insulin for glucose clearance compared to control at week 20 (Figure 2e–h). Changes in glycemic control can be expressed as a product of glucose and insulin, HbA1C (as an indicator of chronic glucose change) or by presenting as a HOMA-IR (homeostatic model of insulin resistance). No significant changes were seen between the groups for HbA1c and HOMA-IR (Appendix A). TheE4 group, however, showed a significant reduction in area under the curves (AUCs) for Glucose×Insulin (Figure 2i–l) after 20 weeks of HF feeding, compared to the control.

### 2.3. E4 and Chow Mice Display Significant Reduction in Serum Triglyceride and Improved Liver Histology

To further examine the effect of HFD on circulating lipids, we determined serum triglyceride levels. In the control group, there was no significant increase in serum triglyceride after an additional 10 weeks of HF-dox diet, compared to that at 10 weeks of HFD (Figure 3a). E4 expression or switching to chow diet from weeks 10–20 both significantly reduced serum triglyceride levels compared to control (Figure 3b,c). The reduction in serum triglyceride levels between week-10 and week 20 was not significantly different in the E4 and chow groups (Appendix A).

To determine hepatic steatosis, formalin fixed liver sections were H&E stained and microscopic scoring was performed by an independent pathologist blinded to the study groups. Observed microscopic changes were graded and summary scores calculated for lesions indicative of macrovesicular steatosis, microvesicular steatosis, hepatocyte hypertrophy and inflammation. A detailed individual score is presented in Table 1. These histological changes were most extensive in the control group compared to both the E4 group and the chow group (Figure 4i). After analyzing the score by using the Wilcoxon rank sum test, compared to control, the chow group displayed significant reductions (*p* < 0.05) in inflammation, steatosis hypertrophy and microvesicular score (Figure 4d,f,h). The E4 group showed significantly lower hepatic hypertrophy (*p* < 0.05) and microvesicular score compared to control (*p* = 0.056) (Figure 4c,e) despite continued HFD. Only mild fibrosis (typically subcapsular) was observed, mostly in control group during the Picrosirius red staining (Figure 4j).

### 2.4. Dietary Fat Restriction Significantly Changes the Expression of Genes for Hepatic Lipid, Glucose, Insulin and Energy Metabolism

To determine molecular signaling changes between chronic HFD feeding, or thereafter switching to dietary fat restriction or E4 protein expression with continued high fat feeding, we analyzed genes involved in hepatic lipid metabolism.

Expressions of the genes involved in fatty acid uptake (plasma membrane fatty acid-binding protein (*Fabppm)*, Long-chain fatty acid transport protein (*Fatp) 1*, *2* and *5)*, denovo lipogenesis (*Fasn*, *Acc1*, *Acc2* and *Chrebp*), intracellular lipid transport (liver fatty acid-binding protein (*L-fabp)* and acyl-CoA-binding protein (*Dbi*), TG synthesis (long-chain acyl-CoA synthetases, *Gpat 1*, *-2* and *-4*), fat oxidation (peroxisome proliferator-activated receptor alpha (*Pparα),* peroxisome proliferator-activated receptor gamma (Pparγ), carnitine palmitoyltransferase 1A *(Cpt1a),* peroxisome proliferator-activated receptor gamma coactivator 1-alpha *(Pgc1α)* and hydroxymethylglutaryl-CoA synthase (*HmgCoAs*)), lipid loading, and droplet formation (Perilipin 5 (*Plin5*)) were all significantly upregulated in chow group compared to control (Figure 5). This suggests no effect of diet reversal on tissue specific expression of genes involved in lipid accumulation even though histological examination did not display increased presence of steatosis.

To evaluate if continued HFD has a similar effect, we compared hepatic expression of genes involved in lipid metabolism following 10 weeks of HFD with genes after 20 weeks of HFD within the control group. No significant differences in hepatic gene expression was observed for lipid metabolism between mice on HFD for 10 weeks and 20 weeks (Appendix A). No significant changes in liver lipid metabolism gene expression were observed in E4 vs control group after 20 weeks of HFD (Figure 5).

The expressions of Fibroblast Growth Factor 21 *(Fgf21)* and phosphoenolpyruvate carboxykinase 1 (*Pepck)* were significantly upregulated only in chow compared to control (Figure 5n). This is contradictory as *Fgf21* expression significantly reduces blood sugar levels and body weights in mice with diet induced obesity by reducing serum triglycerides, reversing hepatic steatosis and suppressing glucose production [27,28], while *Pepck* expression indicates increased gluconeogenesis, which suggests that the body is resisting the weight loss due to diet reversal and trying to increase glucose production. Moreover, *Vegf, and Tgfβ,* were significantly upregulated in the chow group compared with control group (Figure 5i), which are involved in liver tissue maintenance and prevention of fibrosis [29]. Autophagy has a positive role on hepatic health while impairment in autophagy is often correlated with NASH. There were no significant differences for genes involved in autophagy such as autophagy related gene 5 (Atg5), microtubule-associated proteins light chains 3 A or B (Lc3a/Lc3b), beclin1 among the groups (Figure 5o,p). Similarly, there was no significant difference for genes involved in mitochondrial dysfunction, fusion and apopotosis such as BCL2 interacting protein 3 like (Bnip3l), PTEN-induced kinase 1 (Pink1), Mitofusin-1 or 2 (mfn1/mfn2) for all three groups of mice (Figure 5o,p).

### 2.5. Significant Downregulation of Cpt1a in iWAT Suggests Less Fat Oxidation in E4 Mice 

*Cpt1a* expression is linked with mitochondrial entry of fatty acids and oxidation. *Cpt1a* was significantly downregulated in the E4 group compared to control (Figure 6) while there were no significant changes in *Pparα, Pparγ, Pgc1α and* adipose triglyceride lipase *(Atgl).* Interestingly, there was a significant upregulation of *Cpt1a* in control group after twenty weeks of HFD (Figure 6). No changes were found in lipogenic gene expression in iWAT within the groups (Appendix A).

## 3. Discussion

Excess fat accumulation or impaired glycemic control with hyperinsulinemia are some common triggers of NAFLD [30,31]. There are no specific medications for treating NAFLD. Reducing dietary fat intake and losing weight to improve insulin resistance have remained the mainstays of NAFLD treatment [32,33,34]. However, long-term success in reducing both weight and fat intake, is marginal, limiting the impact of NAFLD treatment. Considering the practical difficulties in long-term adherence to lifestyle changes, a therapeutic agent that could facilitate reduction in liver fat independent of dietary fat intake would be highly desirable. Several studies showed improvement in glycemic control by E4 in mice on HFD [16,18,19], along with reduction in endogenous insulin levels, hepatic lipid accumulation and protection from steatosis [16,18]. Here, we aimed to examine the effect of E4 on existing hepatic steatosis. Additionally, we compared the differences, if any, between E4 treatment and reduction in dietary fat intake, in improving HFD-mediated hepatic steatosis.

C57BL6/j background mice used in this study are known to develop severe diet induced obesity, glucose intolerance, moderate insulin resistance and obesity related co-morbidities like diabetes and NAFLD [35,36]. Impaired FA uptake, de novo lipogenesis, intracellular lipid transport, TG synthesis and export, fat oxidation and lipid droplet formation are also associated with hepatic steatosis [37,38]. Downregulation of genes involved in hepatic de novo lipogenesis and intracellular lipid transport and upregulation of genes involved in fat oxidation and TG export are associated with E4 expression [16], though we did not observe these changes in the current study due to the difference in experimental design: we induced steatosis first by exposing mice to HFD, and then expressed E4orf1.

Overall, this study showed that, even in the presence of continued HFD, E4 expression induced many beneficial changes, including attenuation of weight gain, improvement in glycemic control and insulin resistance, reduction in serum triglycerides and, most importantly, histopathological attenuation of hepatic steatosis and inflammation. These changes were comparable to those induced by a reduction in fat intake (chow group).

Despite continued high fat feeding, attenuated weight gain in E4 mice compared to control animals suggests protection from weight gain following E4 induction in adipose tissue [16]. Weight loss is commonly associated with better glycemic control and insulin sensitivity [39,40]. Similar to previous studies, in this study, E4 mice showed insulin sparing action and better glycemic control during GTT [16,18,19,21,41,42]. The unique mode of action of E4, which results in a reduced requirement for endogenous insulin, is due neither to pancreatic beta cell damage, nor to a reduced ability of beta cells to secrete insulin [20,43,44]. The requirement of significantly higher amounts of endogenous insulin after 10 weeks of HF-dox diet, compared to measurements post 10 weeks of HFD in the control group, clearly implies HF induced hyperinsulinemia [45,46]. The continued presence of higher endogenous insulin levels after diet reversal possibly implies that diet change alone is not adequate to attenuate HF induced hyperinsulinemia.

Significant reduction in body weight is often associated with dietary fat restriction [47], however this is not true for all cases [48]. Instead of significant weight loss, sometimes, mice have attenuated weight gains during dietary interventions. “Metabolic memory” and “persistent epigenetic modification” [48,49] are well known contributors in this controversial weight loss theory. To address the aberrant upregulation of hepatic lipogenic genes after a diet change from HF to chow, we need to account for the HFD-induced disruption of daily rhythm, chromatin modification, carbohydrate to fat ratio in the diet, genetic modification/strain of the mice and overall “metabolic memory” of the cells [47,48,49,50]. There are contradictory studies reporting both reversal of HF-induced lipogenic genes to normal (in chow fed mice) upon diet change and vice-versa. In these studies, ongoing transcriptional changes promoted by HFD were identified by RNA-seq analysis. Siersbaek et al. suggested complete reversal for most of the HFD induced hepatic genes as well as body weight after introduction of chow diet [47]. On the other hand, Leung et al. showed persistent chromatin modification in the liver induced by HFD in C57BL/6J and A/J mice [48]. Interestingly, attenuated weight gain after diet change (not weight reduction) was observed as well as elevated level of hepatic TG and lipid storage in both HFD and dietary fat restricted mice compared to control [48,51]. Additionally, in DIO mice, switching to very-low carbohydrate (VLCD) or low-fat isoenergetic diet (LFD) showed similar weight loss but VLCD-fed mice showed increased mRNA levels of hepatic lipogenesis [52].

Therefore, it appears that two things might be occurring in the mice switched to chow diet. Due to diet reversal, the mice are in negative energy balance as observed by weight loss and upregulation of markers of fat oxidation, and to maintain its energy balance the body has to increase lipid stores. In humans, to compensate for negative energy balance, the body tries to resist weight loss by stepping up hunger, reducing satiety, increasing fatty acid uptake, lipogenesis, TG synthesis or lipid droplet formation [53].

Glucose and fatty acids have reciprocal relationship in terms of metabolism [54]. In a fasting condition, increased FA oxidation inhibits glucose oxidation and promotes gluconeogenesis to maintain the blood glucose level. In a fed state, or upon availability of glucose, there is an increase in glucose oxidation and lipid storage, thereby inhibiting FA oxidation [55]. Significant downregulation of *Cpt1a* in iWAT of E4 implies reduced fat oxidation and supports enhanced glucose uptake by the tissue, though more research is warranted. Overall expression of *Cpt1a* is very low in iWAT compared to muscle and liver [56]. This establishes that excessive up or down regulation of *Cpt1a* in iWAT has minimal effect on the overall energy expenditure of the body.

In conclusion, despite 60% HFD for twenty weeks, the E4 group evinced attenuated weight gain, and a reduced requirement for endogenous insulin and serum TG. Weight loss and reduction in serum TG were also observed after dietary fat restriction in chow mice. Both groups showed less steatosis in the liver compared to control mice, while chow mice were on HFD only for ten weeks and E4 mice were on HFD for twenty weeks. Therefore, the present study confirms that both E4 expression and dietary fat restriction have a beneficial effect in treating HFD-induced hepatic steatosis. Significant upregulation of many metabolically important genes in spite of diet-induced weight loss, however, suggests that there is a continued risk of weight regain and lipid accumulation [57]. Maintaining long term weight loss and lifestyle change is challenging, and therefore, E4orf1 appears to be potentially a better candidate than weight loss alone for treating hepatic steatosis and for preventing its progression to NASH.

## 4. Materials and Methods

### 4.1. Animal Study

Fourteen to eighteen-week-old E4 transgenic mice of C57BL6/J background (n = 24; male and female) were placed on a 60% HFD (58Y1, TestDiet) for ten weeks. After ten weeks, n = 4 mice were sacrificed to determine hepatic lipid accumulation following high fat feeding, while the remaining n = 20 mice were randomly divided into two groups for an additional ten weeks. One group (n = 6 mice; n = 3 males and n = 3 females) received regular chow diet (5V5R, TestDiet), and the other group (n = 14 mice; n = 6 males and n = 8 females) received a 60% HFD supplemented with doxycycline (5G3T, TestDiet) to induce E4orf1 protein expression in the adipose tissue. All animals were housed in the same room, ≤5 mice/cage on a 12-h light/12-h dark cycle and received humane care. The experimental procedures were approved by Institutional Animal Care and Use Committee (IACUC) at Texas Tech University, Lubbock, TX. Mice had access to ad-libitum diet and water throughout the study and were weighed weekly using a digital scale (OHAUS Model CL 201). Presence of E4orf1 transgene in these mice was confirmed by DNA genotyping collected via tail snip. However, presence of the transgene does not confirm E4orf1 protein expression, hence, adipose tissue depots from all mice were screened for E4orf1 protein expression. The mice were divided into E4 group (E4orf1 expressing mice; n = 7; n = 3 males and n = 4 females) and control group (E4orf1 non-expressing mice; n = 7; n = 4 males and n = 3 females).

### 4.2. Tissue Collection and Storage

At termination of the study, mice were euthanized using CO_2_ asphyxiation and cervical dislocation. Trunk blood was collected in non-heparinized tubes and blood serum obtained after spinning at 14,000 rpm for 15 min (18,620× *g*) at 4 °C and stored at −80 °C. Liver and adipose tissues were collected during necropsy and either flash frozen in liquid nitrogen, or stored in RNALater (Invitrogen, cat. no. AM7020) or fixed in 10% neutral buffered formalin (NBF) for protein, RNA, and histology analysis, respectively.

### 4.3. RNA Extraction

Total RNA was extracted from the liver and adipose tissue using RNeasy^®^ Plus Universal Mini Kit (QIAGEN, Hilden, Germany, cat. no. 73404). Tissue samples of each mouse were collected in 2 mL tubes and dissolved (~20 mg/900 μL) in QIAzol^®^ reagent (Qiagen Sciences, Frederick, MA, USA, cat. no. 79306). Two 0.45 μm steel beads were added to each tube and homogenized in Tissuelyser LT for 5 min at 50 Hz. RNA extraction kit manufacturer’s instructions were followed for rest of the procedures.

### 4.4. cDNA Preparation and qRT-PCR

Complementary DNA (cDNA) was synthesized using the Maxima H Minus First Strand Complementary DNA Synthesis Kit (Thermo Fisher Scientific, Waltham, MA, USA, cat. no. K1681) with 1 μg of RNA.

The expression levels of genes associated with liver and adipose tissue lipid metabolism were determined by quantitative real-time polymerase chain reaction (qRT-PCR). Specific primers for each gene (listed in Table 2) were designed using Sigma Aldrich Oligo Architect software. The RT-PCR reaction mix had a final volume of 20 μL; 25 ng of cDNA, 450 nM of the forward and reverse primers, and 10 μL of 1X SsoAdvanced™ Universal SYBR^®^ Green Supermix (Bio-Rad Laboratories, cat. no. 172-5271). PCR reactions were carried out in 96-well plates using the Bio-Rad CFX RT-PCR detection system. Mouse *β actin* gene was used as the reference for liver and adipose tissue.

### 4.5. Serum Triglyceride Measurement

Serum TG (mg/L) were measured using the Cayman kit (cat. no. 10010303). Briefly, 10 µL of serum sample was added to 2 wells in a 96-wells polystyrene microplate and mixed with 150 µL of enzyme solution (ingredients were supplied with the kit) with multichannel pipette. The plate was incubated for 15 min at room temperature. Absorbances for both sample and standard were measured at 540 nm.

### 4.6. Hemoglobin A1c (HbA1c) Measurement in Blood

HbA1c was measured using Crystal chem mouse HbA1c kit (cat. no. 80310). Briefly, 5 µL of blood was taken and mixed with lysate buffer to prepare sample lysate. Next, 25 µL of sample lysates were added to 2 wells in a 96-wells polystyrene microplate and mixed with protease buffer A and buffer 1B. The plate was incubated for 5 min at 37 °C and the absorbance was measured at 700 nm. Next, enzyme solution was added and incubated for 3 min at 37 °C and the final absorbance was measured at 700 nm. The difference between the two absorbance measurements was used for calculating HbA1c.

### 4.7. Glucose Tolerance Test

Glucose tolerance test (GTT) was performed after ten weeks of HFD and at the end of the study (after 20 weeks). GTT involved an intraperitoneal injection of glucose (2 g/kg body weight) after a 4 h fast. During ipGTT, serum glucose levels were measured by tail bleed at 0 min and following glucose load at 15, 30, 60, and 120 min. Blood glucose (mg/dL) during GTT was measured using AlphaTrak2 glucose meter. 20 μL Blood was also collected at these time points in heparinized microvette (Sarstedt, cat. no. 16.444.100) for insulin measurement.

### 4.8. Serum Insulin Measurement

Blood samples were collected during GTT in EDTA-coated microvettes (Sarstedt, cat. 16.444.100) and processed by centrifuging at 10,000 rpm for 10 min, and the serum was collected for further analysis. Serum insulin (ng/mL) was measured using the ELISA kit (EMD Millipore, cat. no. EZRMI-13K). The manufacturer’s instructions were followed throughout the analysis. The highest amount of insulin the assay could measure was 10 ng/mL. Considering insulin levels increase after glucose load and development of hyperinsulinemia after HF diet, serum samples were diluted accordingly to be used in the assay. For standard curve preparation, 4-parameter logistic curve fit was used using GraphPad Prism software (version 9.3.1 (350)).

### 4.9. Liver Histology

Histological analysis of 10% NBF fixed liver samples from five mice from each group, control, E4 and chow, were performed by IDEXX BioAnalytics, Columbia, MO. An H&E slide and a Picrosirius red slide (each with 1 section of liver) were made of the fixed liver samples for each mouse. Microscopic scoring of the liver was performed by a pathologist blinded to study groups. Observed microscopic changes were graded as to severity utilizing the grading system for rodent NAFLD [58]. The one exception included evaluation of 10 fields to count the number of inflammatory foci, instead of 5 fields as was indicated in the reference system. The average of the 10 fields was used to provide a single inflammation score (0–3) for each mouse. In a few samples, only 8 fields were counted due to smaller tissue sample size, and the average of the 8 fields was used to provide a single inflammation score. Summary scores for the liver were calculated for lesions indicative of NAFLD/NASH, including macro-vesicular steatosis, micro-vesicular steatosis, hepatocyte hypertrophy, and inflammation. Fibrosis was also scored on Picrosirius red stained sections utilizing a standard grading system whereby 0 = no significant fibrosis, 1 = mild fibrosis, 2 = moderate fibrosis, and 3 = massive fibrosis. Summary scores did not include fibrosis scoring.

### 4.10. Statistical Analyses

All results are presented as mean ± standard error of the mean. Comparison between the two groups was calculated using Welch’s *t*-test assuming unequal variance in the values for each group. For GTT glucose and insulin data, the area under the curve (AUC) was calculated. In addition, to account for the effect of time and treatment on GTT glucose and insulin data, two-way repeated measure analysis of variance was used. The HOMA-IR value was calculated using equation HOMA-IR = Fasting blood glucose (mg/dL) × Fasting insulin (ng/mL) × 0.072. The relative amount of all mRNAs was calculated using the 2^−^^ΔΔCT^ method. For histological data, a two-sample Wilcoxon test (rank sum test) was performed.

## Figures and Tables

**Figure 1 ijms-23-09286-f001:**
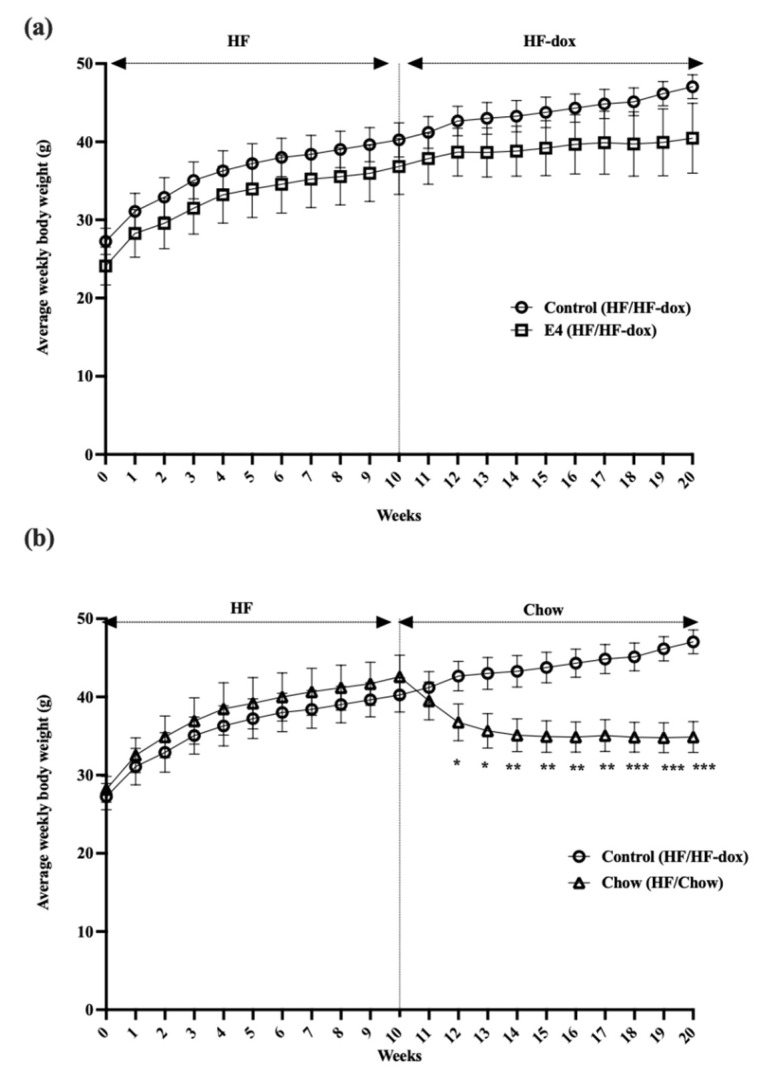
Changes in average weekly body weight. (**a**) Average weekly body weights in control, E4 and chow mice. Control mice show significant weight gain on HFD at the end of the study (week 20) compared with baseline (week 0), while E4 expression from week 10 attenuated weight gain among the E4 mice during the additional 10 weeks of HFD. (**b**) Average weekly body weight changes in chow mice following diet change, compared with control mice. Chow mice show significant weight loss following diet reversal at week 10 compared with control mice. n = 5; Welch’s *t*-test: * *p* < 0.05, ** *p* < 0.01, *** *p* < 0.001.

**Figure 2 ijms-23-09286-f002:**
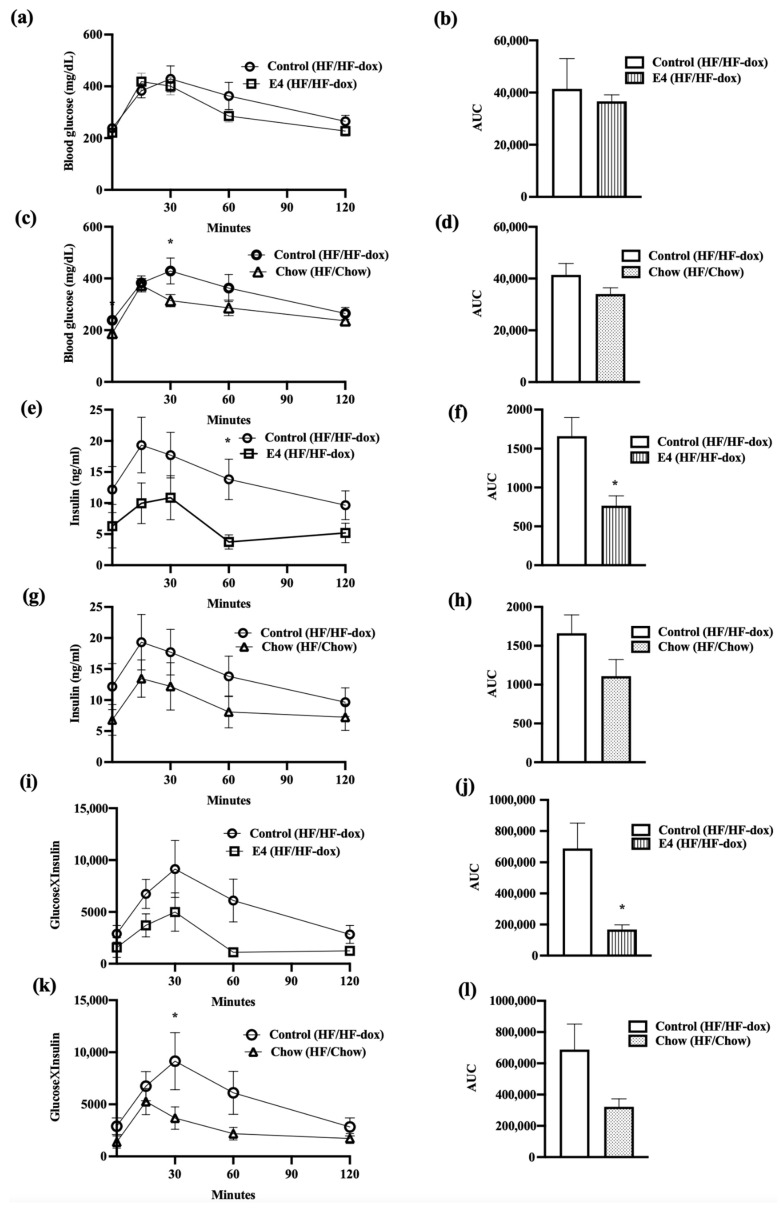
Changes in glycemic control and subsequent insulin requirement during glucose tolerance test (GTT). (**a**,**b**) represents glucose clearance and corresponding AUC in control and E4. (**c**,**d**) represents glucose clearance and corresponding AUC in control and chow groups. (**e**–**h**) represents endogenous insulin response to glucose load and corresponding AUCs in control vs. E4 and control vs chow groups, respectively. (**i**,**j**) compares product of glucose and insulin (glucose × insulin) and relative AUCs between control and E4 while (**k**,**l**) represents same for control vs. chow. n = 5; Welch’s *t*-test: * *p* < 0.05.

**Figure 3 ijms-23-09286-f003:**
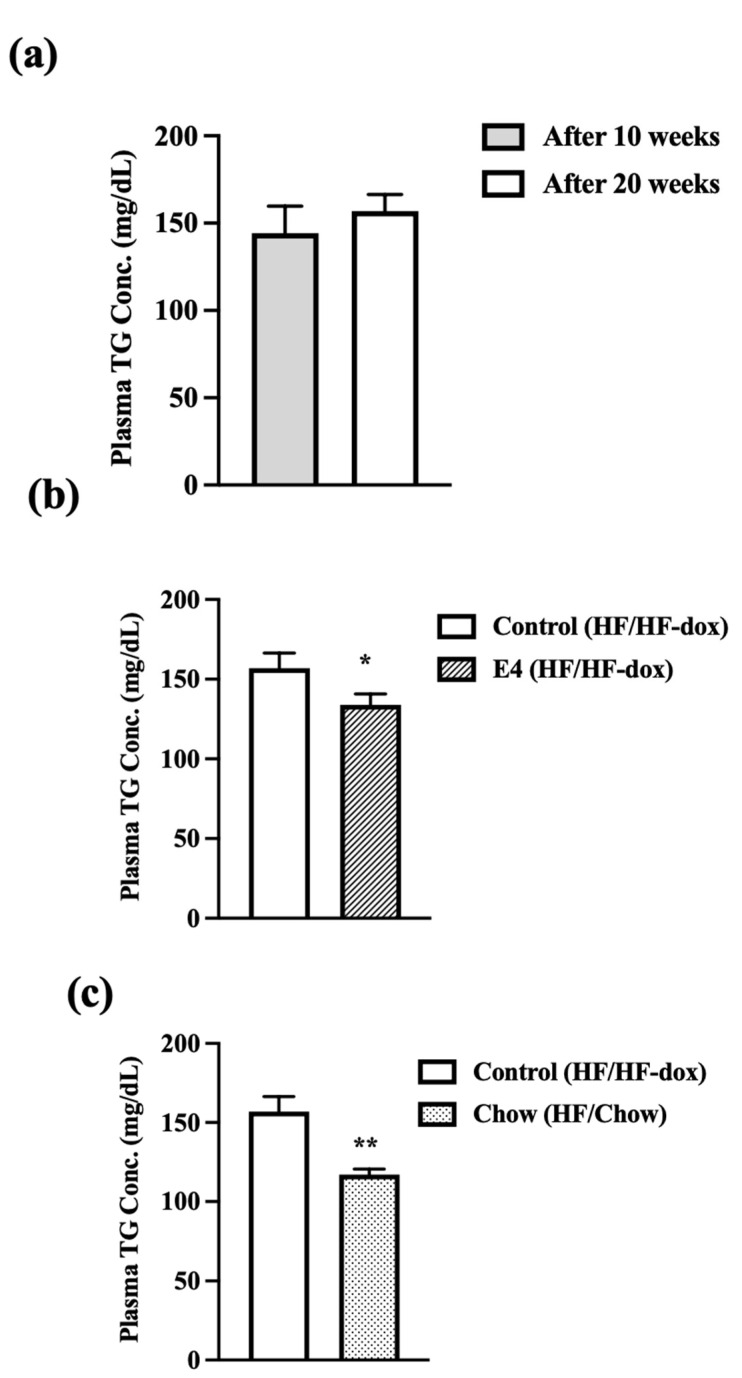
Plasma TG levels were measured after 10 and 20 weeks of high fat feeding in E4 and control mice and 10 weeks of diet reversal in chow mice. (**a**) Control mice do show any increase in plasma TG concentration after 20 weeks of HFD compared with post 10 weeks of HFD. Both (**b**) E4 and (**c**) chow group show significant reduction in serum TG content compared to control at 20 weeks. n = 5, Welch’s *t*-test: * *p* < 0.05, ** *p* < 0.01.

**Figure 4 ijms-23-09286-f004:**
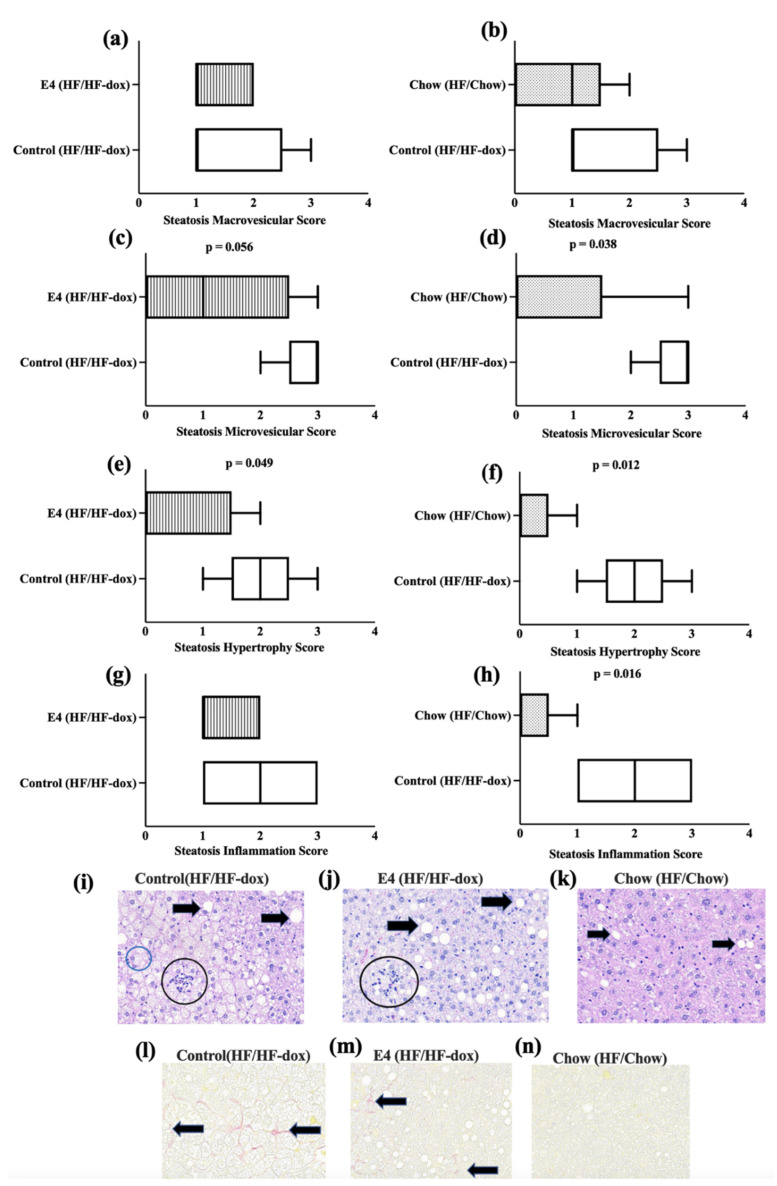
Pathological changes in the liver tissue. (**a**–**h**) Wilcoxon rank sum analysis for hepatic pathological scores among the groups. (**i**–**k**) represent H&E staining of liver tissue from control, E4 and chow mice respectively. Black arrows indicate macrovesicular and microsvesicular steatosis. Black circles indicate cellular infiltrates (inflammation) and blue circle indicates cellular hypertrophy. (**l**–**n**) represent pico sirius red staining of liver tissue from control, E4 and chow mice respectively. Black arrows indicate fibrosis. n = 5.

**Figure 5 ijms-23-09286-f005:**
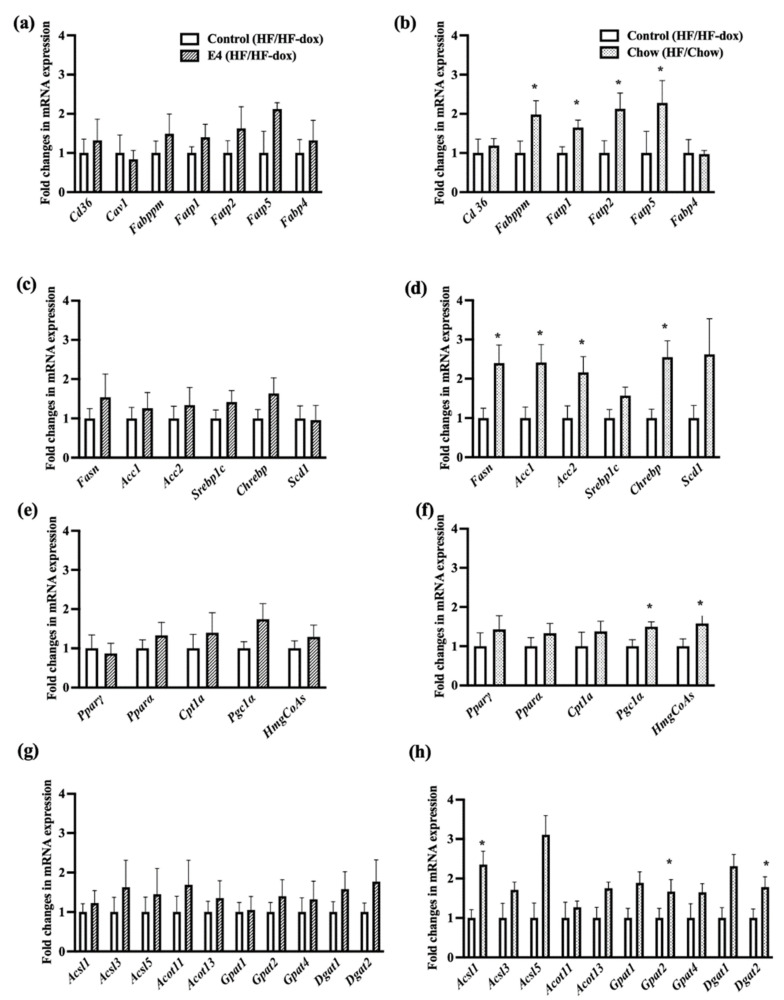
Fold changes in mRNA expression in liver tissue. The left panel is for control vs E4 mice and the right panel is for control vs chow mice. Genes involved in fatty acid uptake (**a**,**b**), de novo lipogenesis (**c**,**d**), Fat oxidation (**e**,**f**), TG synthesis (**g**,**h**), intracellular lipid transport, lipid loading and droplet formation (**i**,**j**), fibrosis and inflammation (**k**,**l**), glucose and insulin metabolism (**m**,**n**), mitochondrial dysfunction, autophagy and apoptosis (**o**,**p**). n = 5 Welch’s *t*-test: * *p* < 0.05.

**Figure 6 ijms-23-09286-f006:**
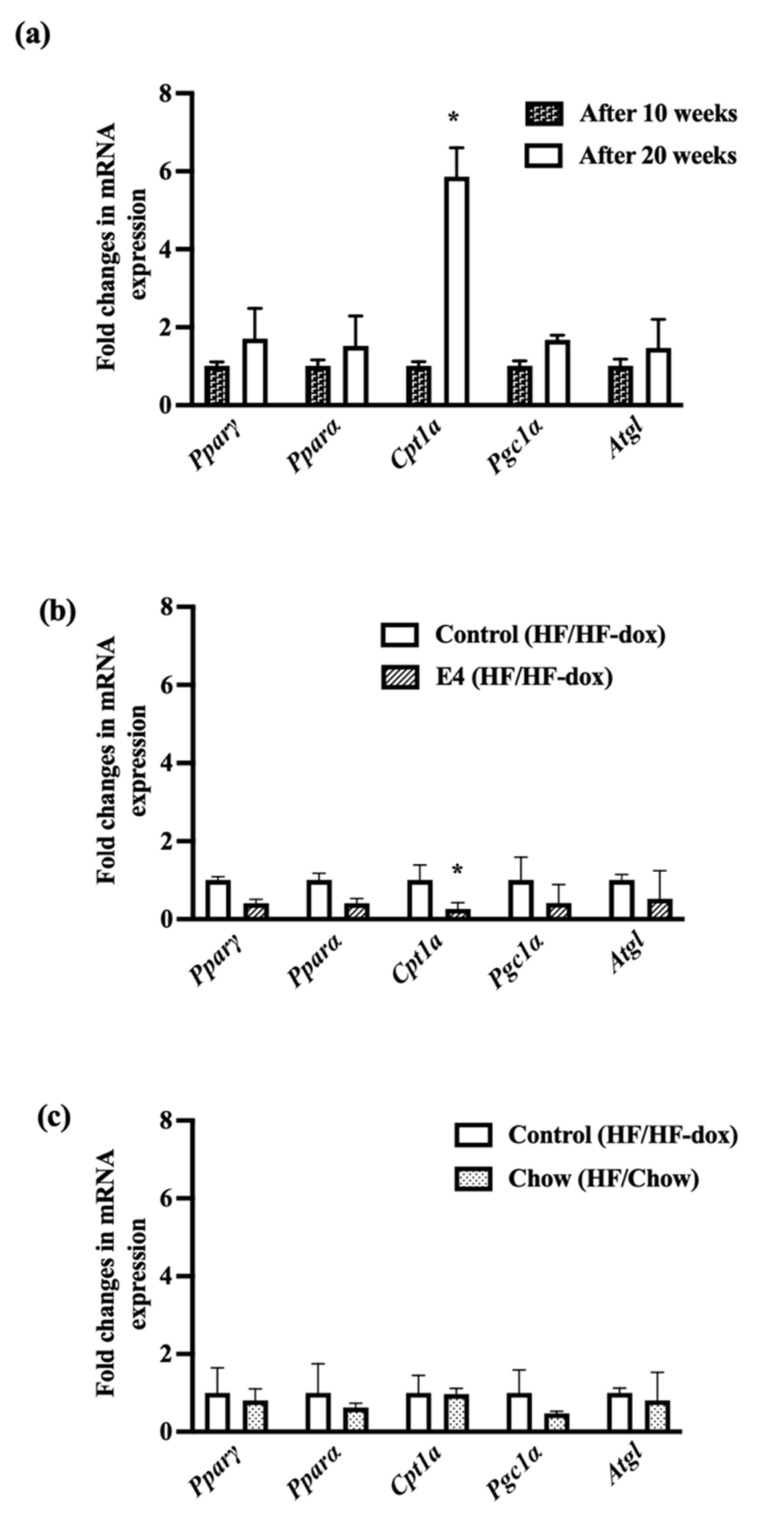
Fold changes in mRNA expression of genes involved in fat oxidation and lipolysis. Control mice show significantly increased expression of Cpt1a from 10 weeks to 20 weeks of HFD feeding (**a**); however, E4 mice show significantly reduced expression of Cpt1a (**b**), while diet reversal in chow mice showed no difference in fat oxidation (**c**). n = 5 Welch’s *t*-test: * *p* < 0.05.

**Table 1 ijms-23-09286-t001:** Pathological score for histological hepatic tissue changes. Summary scores for the liver were calculated for lesions indicative of NAFLD/NASH, including macro-vesicular steatosis, micro-vesicular steatosis, hepatocyte hypertrophy, and inflammation as described in “Materials and Methods”.

Control							
Sex	F	M	M	M	F		
N	5						
Sample ID	L6	L7	L8	L9	L10	# Abnormal	Average
Steatosis: macrovesicular	2	1	1	1	3	5	1.60
Steatosis: microvesicular	2	3	3	3	3	5	2.80
Steatosis: hypertrophy	1	3	2	2	2	5	2.00
Inflammation	2	3	1	1	3	5	2.00
Sum-Scores:	7	10	7	7	11		8.40
Fibrosis	0	1	0	1	0	2	0.40
**E4**							
**Sex**	**M**	**F**	**F**	**M**	**F**		
**N**	**5**						
**Sample ID**	**L1**	**L2**	**L3**	**L4**	**L5**	**# Abnormal**	**Average**
Steatosis: macrovesicular	2	1	2	1	1	5	1.40
Steatosis: microvesicular	3	1	2	0	0	3	1.20
Steatosis: hypertrophy	2	0	1	0	0	2	0.60
Inflammation	1	2	2	1	1	5	1.40
Sum-Scores:	8	4	7	2	2		4.60
Fibrosis	1	1	1	1	0	4	0.80
**Chow**							
**Sex**	**M**	**M**	**M**	**F**	**F**		
**N**	**5**						
**Sample ID**	**L11**	**L12**	**L13**	**L14**	**L15**	**# Abnormal**	**Average**
Steatosis: macrovesicular	1	2	1	0	0	3	0.80
Steatosis: microvesicular	0	3	0	0	0	1	0.60
Steatosis: hypertrophy	0	1	0	0	0	0	0.20
Inflammation	0	1	0	0	0	1	0.20
Sum-Scores:	1	7	1	0	0		1.80
Fibrosis	1	1	0	0	0	0	0.40

**Table 2 ijms-23-09286-t002:** Primer sequences for genes used in RT-qPCR.

Gene Name	Primer Sequence (5′-3′)
*ACC1-Fw*	GCAGCAGTTACACCACATAC
*ACC1-Rev*	TCCGCCATCTTCCACAATA
*ACC2-Fw*	TACGGCGGCATCAAGTAT
*ACC2-Rev*	ACTGTCAACCTCTTCCTTCAT
*ACOT11-Fw*	CTGACTCTTGGCTCTACTTGT
*ACOT11-Rev*	CTCTGAACCTCCGCTCTC
*ACOT13-Fw*	ACGAGAAGTAATGAAGGTTATGTT
*ACOT13-Rev*	AGATGCTGTCCACTAAGGT
*ACSL1-Fw*	CAACACTGAAGGCGAAGAG
*ACSL1-Rev*	CGAGGAGGATTGTGGAGAT
*ACSL3-Fw*	AGGAAGATGTGTATATTGGCTACT
*ACSL3-Rev*	CTGCTAATGTCTGTGGTGAAG
*ACSL5-Fw*	CCATCTCCACTCCAGTCTT
*ACSL5-Rev*	TGTCAGCCACATCTTCCA
*ATG5-Fw*	TCCATCCAAGGATGCGGTTG
*ATG5-Rev*	TCTGCATTTCGTTGATCACTTGAC
*ATGL Fw*	CGCTATGATGGCAATGTGTAT
*ATGL Rev*	TAGTAAGATTCGTGGACCTCTG
*Beclin1-Fw*	ACCAGCGGGAGTATAGTGAGT
*Beclin1-Rev*	CAGCTGGATCTGGGCGTAG
*beta-actin Fw*	AATCTTCCGCCTTAATACTTCATT
*beta-actin Rev*	CTGCCTCAACACCTCAAC
*BniP3i-Fw*	GCACGTTCCTTCCTCGTCT
*BniP3i-Rev*	GCTCTGTCCCGACTCATGC
*CD36-Fw*	AGGTCTATCTACGCTGTGTTC
*CD36-Rev*	AGGCATTGGCTGGAAGAA
*ChREBP1-Fw*	TTCCACAAGCATCCTGACT
*ChREBP1-Rev*	AGAAGCGTGTTCACAAGTTG
*Collagen type IV Fw*	CCAGAGGAGGTGTATAGATAGC
*Collagen type IV Rev*	GCAGAGCAGAAGCAAGAAG
*Collagen14a-Fw*	ATCCTCTATGCTCCTCTC
*Collagen14a-Rev*	CCACTCAGTTCAATGTCT
*Collagen1a-Fw*	AGGTATGCTTGATCTGTAT
*Collagen1a-Rev*	CAGTCCAGTTCTTCATTG
*Collagen3a-Fw*	CAACGGTCATACTCATTC
*Collagen3a-Rev*	TATAGTCTTCAGGTCTCAG
*CPT1a-Fw*	CAAGCCAGACGAAGAACATC
*CPT1a-Rev*	TGACCATAGCCATCCAGATT
*DGAT1-Fw*	GATTGGTGGAATGCTGAGTC
*DGAT1-Rev*	GGCTTGTAGAAGTGTCTGATG
*DGAT2-Fw*	TCCAGAAGAAGTTCCAGAAGTAT
*DGAT2-Rev*	CAGGTGTCAGAGGAGAAGAG
*FABP4-Fw*	TGGACTTCAGAGGCTCATAG
*FABP4-Rev*	CCACAAAGGCATCACACAT
*FABPpm-Fw*	CGAGCAGTGGAAGGAGAT
*FABPpm-Rev*	GCAGAGGCAGACATTGATG
*Fasn-Fw*	GTCGTCTATACCACTGCTTACT
*Fasn-Rev*	ACACCACCTGAACCTGAG
*FATP1-Fw*	AGACTCAGGAAGGTTGTTGT
*FATP1-Rev*	AGATGAAGGCAGGCAGAG
*FATP2-Rev*	CCATACACATTCACTTCTTCAACA
*FATP2-Fw*	AGGCGACATCTACTTCAACA
*FATP5-Fw*	AGCCAGCCATCTTATCACAT
*FATP5-Rev*	AAGCAGCCAAGGAATCCA
*FGF21-Fw*	CCAAGACCAAGCAGGATTC
*FGF21-Rev*	AGAGTCAGGACGCATAGC
*Fibronectin-Fw*	GGTTGATGATACTTCCATTGTTGT
*Fibronectin-Rev*	GTGCTACTGCCTTCTACTGA
*Foxo1-Fw*	GCTCTGTCCTGAAGAATCCT
*Foxo1-Rev*	CTAATCCTGCCACTGTCTGTA
*G6Pase-Fw*	GGAAGGATGGAGGAAGGAAT
*G6Pase-Rev*	TCAGGTCAGCAATCACAGA
*GPAT1-Fw*	CTATCCAGTAACGAGTCCAGAA
*GPAT1-Rev*	GGCGGTGAAGAGAATGTG
*GPAT2-Fw*	GTCTTCCTACTGCTACTGTCA
*GPAT2-Rev*	TGCTGTCTTCCTGTGTCA
*GPAT4-Fw*	GGTGGAGAACAGCGAGTA
*GPAT4-Rev*	TCAGAAGGAAGGACAGAAGG
*HMGCS2-Fw*	CTTGAACGAGTGGATGAGATG
*HMGCS2-Rev*	CTATGAGGCTGCTGTGTCTA
*IL-18 Fw*	CCAAGTTCTCTTCGTTGACAA
*IL-18 Rev*	TCACAGCCAGTCCTCTTAC
*IL-1β-Fw*	TTCAGGCAGGCAGTATCA
*IL-1β-Rev*	CCAGCAGGTTATCATCATCATC
*IL-6-Fw*	ACAGAAGGAGTGGCTAAG
*IL-6-Rev*	AGAGAACAACATAAGTCAGATAC
*Lc3a-Fw*	CCCATCGCTGACATCTATGAAC
*Lc3a-Rev*	AAGGTTTCTTGGGAGGCGTA
*Lc3b-Fw*	TCCACTCCCATCTCCGAAGT
*Lc3b-Rev*	TTGCTGTCCCGAATGTCTCC
*Lipin1-Fw*	GCCGTGTCATATCAGCAAT
*Lipin1-Rev*	ATCGCCAGAAGTAGAGGAG
*MCP1-Fw*	GCCCCACTCACCTGCTGCTACT
*MCP1-Rev*	CCTGCTGCTGGTGATCCTCTTGT
*mFn1-Fw*	GCAGACAGCACATGGAGAGA
*mFn1-Rev*	GATCCGATTCCGAGCTTCCG
*mFn2-Fw*	TGCACCGCCATATAGAGGAAG
*mFn2-Rev*	TCTGCAGTGAACTGGCAATG
*PEPCK-Fw*	GACATTGCCTGGATGAAGTT
*PEPCK-Rev*	CGTTGGTGAAGATGGTGTT
*PGC1a-Fw*	ACAATAACAACAACAACCATACCA
*PGC1a-Rev*	ATTCTGTCTCTTGCCTCTTCA
*Pink1-Fw*	CCATCGGGATCTCAAGTCCG
*Pink1-Rev*	GATCACTAGCCAGGGACAGC
*PLIN2-Fw*	AGAAGCCGAGCAACTATGA
*PLIN2-Rev*	TGAGAGCCTGGTGATAAGC
*PLIN3-Fw*	CGACAGGAGCAGAACTACT
*PLIN3-Rev*	CCGAGCACACTTGTTAGC
*PLIN5-Fw*	CACAGTGGAGGAGCAGAG
*PLIN5-Rev*	AAGAGTGTTCATAGGCGAGAT
*PPARα-Fw*	TCGCTATCCAGGCAGAAG
*PPARα-Rev*	ACAACAACAACAATAACCACAGA
*PPARγ-Fw*	CCACCAACTTCGGAATCAG
*PPARγ-Rev*	GCTCTTGTGAATGGAATGTCT
*SCD1-Fw*	TGCCTCTTAGCCACTGAAT
*SCD1-Rev*	ACTGTTGAGATGTGAGACTGT
*Srebp1c-Fw*	GCTTCTCTTCTGCTTCTCTG
*Srebp1c-Rev*	GGCTGTAGGATGGTGAGT
*TGFβ-Rev*	AAGGTAACGCCAGGAATTG
*TGFβ-Fw*	GCAACAACGCCATCTATGA
*VEGF A Fw*	CTCTTCTCGCTCCGTAGTAG
*VEGF A Rev*	CCTCTCCTCTTCCTTCTCTTC

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
