# Peer review of "E4orf1 Prevents Progression of Fatty Liver Disease in Mice on High Fat Diet"

_ijms, 2022, doi:10.3390/ijms23169286_

Round 1

Reviewer 1 Report

In the manuscript “E4orf1 Prevents Progression of Fatty Liver Disease in Mice on 2 High Fat Diet”, Afruza and colleagues studied the impact of E4orf1, a protein from the human adenovirus Ad36 which is responsible for its anti-hyperglycemic effect. The paper is interesting and relevant. Nevertheless, I have some suggestions that could improve the scientific outcome of the manuscript.

The authors mention in the abstract that "Non-alcoholic fatty liver disease (NAFLD) is a consequence of hepatic steatosis.". However, simple steatosis (non-alcoholic fatty liver; NAFL) is already the first stage of NAFLD.

Regarding insulin measurements, blood was taken in 5 timepoints of the ipGTT. How much volume was taken each time? Was the volemia of this animals assured?

The authors state that the animals needed to secrete more insulin to deal with hyperglycemia. However, and given that insulin clearance occurs mostly in the liver due to the action of CEACAM1 and IDE (PMID: 30968756; PMID: 33631143), did the authors considered to measure C-Peptide?

I would prefer to see the three groups in the same graphs instead of separating them, because the paper becomes more confusing.

In Fig 2b and 2d, the values of the control group are different, and the Chow and E4 groups seem to be equal. Is this correct?

Minor comments:

Line 305: "mice" is repeated

Line 332: is "0.45um beads" correct?

Author Response

Comment 1. The authors mention in the abstract that "Non-alcoholic fatty liver disease (NAFLD) is a consequence of hepatic steatosis.". However, simple steatosis (non-alcoholic fatty liver; NAFL) is already the first stage of NAFLD. 

Response 1: We have re-phrased the sentence to “Non-alcoholic fatty liver disease (NAFLD) covers a broad spectrum of liver diseases ranging from steatosis to cirrhosis.”

 Comment 2. Regarding insulin measurements, blood was taken in 5 timepoints of the ipGTT. How much volume was taken each time? Was the volemia of this animals assured? 

 Response 2: For each time point ~40-50 ul blood was collected. At baseline, the mice had a body weight of  ~25-28g and at 10 and 20 weeks their body weights were ~35-40g. Considering the maximum amount of blood that can be drawn from a rodent in any 2-week period may not exceed 1% of the animal's body weight.  For a mouse weighing 25 grams, the maximum allowable blood collection may not exceed 0.25 grams or 0.25 ml. Furthermore, immediately following the GTT, mice were returned to their original cages with ad libitum food and water. Therefore, the volemia for these mice was assured.

 Comment 3. The authors state that the animals needed to secrete more insulin to deal with hyperglycemia. However, and given that insulin clearance occurs mostly in the liver due to the action of CEACAM1 and IDE (PMID: 30968756; PMID: 33631143), did the authors considered to measure C-Peptide? 

 Response 3: We agree with the reviewer that insulin clearance occurs in the liver. Our goal in this study was measure insulin secretion from the pancreas in response to a glucose bolus during ipGTT, therefore, in this study we did not measure C-peptide.

 Comment 4. I would prefer to see the three groups in the same graphs instead of separating them, because the paper becomes more confusing.

Response 4: As requested by the reviewer, we tried to recreate the graphs combining all three groups. The clarity in the graphs was lost and we would require to change statistical analysis from T-Test to ANOVA, which would further require us to re-write the results and discussion. Therefore, we respectfully request the reviewer and the editor to please allow the results as currently presented.

 Comment 5. In Fig 2b and 2d, the values of the control group are different, and the Chow and E4 groups seem to be equal. Is this correct?

 Response 5: The average AUC values for control=41700, E4=37903 and chow=33525. Therefore, there was no statistically significant difference between control and E4 or control and chow.

Minor comments:

Comment 6. Line 305: "mice" is repeated 

Response 6: This has been corrected.

Comment 7. Line 332: is "0.45um beads" correct? 

Response 7: This has been corrected.

Reviewer 2 Report

The authors of the article are thanked for the quality of the work and the choice of the subject relating to the prevention of hepatic steatosis or its progression to liver disease. It is known that  Non-Alcoholic Fatty Liver Disease (NAFLD) is one of the most common liver disorders worldwide. Afruza and collaborators discuss if transgenic (Tg) doxycycline-induced expression in adipose tissue of E4orf1 (E4), an adenoviral 10 protein, or dietary fat restriction attenuated hepatic steatosis or its progression in mice. The manuscript entitled “E4orf1 Prevents Progression of Fatty Liver Disease in Mice on High Fat Diet” is a good study, scientifically valid, well executed, and deserve some space in the journal. After reading the manuscript thoroughly, I have no comments to the Authors. I believe the manuscript is very good and can be published in present form.

Author Response

Response: We thank the reviewer for the encouraging and positive comments.